# Uropathogens’ Antibiotic Resistance Evolution in a Female Population: A Sequential Multi-Year Comparative Analysis

**DOI:** 10.3390/antibiotics12060948

**Published:** 2023-05-23

**Authors:** Cristian Mareș, Răzvan-Cosmin Petca, Răzvan-Ionuț Popescu, Aida Petca, Bogdan Florin Geavlete, Viorel Jinga

**Affiliations:** 1Department of Urology, “Carol Davila” University of Medicine and Pharmacy, 8 Eroii Sanitari Blvd., 050474 Bucharest, Romania; cristian.mares@drd.umfcd.ro (C.M.); razvan-ionut.popescu@drd.umfcd.ro (R.-I.P.); bogdan.geavlete@umfcd.ro (B.F.G.); viorel.jinga@umfcd.ro (V.J.); 2Department of Urology, “Saint John” Clinical Emergency Hospital, 13 Vitan-Barzesti Str., 042122 Bucharest, Romania; 3Department of Urology, “Prof. Dr. Th. Burghele” Clinical Hospital, 20 Panduri Str., 050659 Bucharest, Romania; 4Department of Obstetrics and Gynecology, “Carol Davila” University of Medicine and Pharmacy, 8 Eroii Sanitari Blvd., 050474 Bucharest, Romania; aida.petca@umfcd.ro; 5Department of Obstetrics and Gynecology, Elias University Emergency Hospital, 17 Marasti Blvd., 011461 Bucharest, Romania; 6Medical Sciences Section, Academy of Romanian Scientists, 050085 Bucharest, Romania

**Keywords:** UTIs, females, AMR, uropathogens, antibiotic resistance, *Escherichia coli*, *Klebsiella*, COVID-19

## Abstract

Urinary Tract Infections (UTIs) represent a common finding among females and an important basis for antibiotic treatment. Considering the significant increase in antibiotic resistance during the last decades, this study retrospectively follows the incidence of uropathogens and the evolution of resistance rates in the short and medium term. The current study was conducted at the “Prof. Dr. Th. Burghele” Clinical Hospital, including 1124 positive urine cultures, in three periods of four months between 2018 and 2022. *Escherichia coli* was the most frequent uropathogen (54.53%), followed by *Klebsiella* spp. (16.54%), and *Enterococcus* spp. (14.59%). The incidence of UTIs among the female population is directly proportional to age, with few exceptions. The highest overall resistance in Gram-negative uropathogens was observed for levofloxacin 30.69%, followed by ceftazidime 13.77% and amikacin 9.86%. The highest resistance in Gram-positive uropathogens was observed for levofloxacin 2018-R = 34.34%, 2020-R = 50.0%, and 2022-R = 44.92%, and penicillin 2018-R = 36.36%, 2020-R = 41.17%, and 2022-R = 37.68%. In Gram-negative uropathogens, a linear evolution was observed for ceftazidime 2018-R = 11.08%, 2020-R = 13.58%, and 2022-R = 17.33%, and levofloxacin 2018-R = 28.45%, 2020-R = 33.33%, and 2022-R = 35.0%. The current knowledge dictates the need to continuously assess antimicrobial resistance patterns, information that is necessary for treatment recommendations. The present study aims to determine the current situation and the evolution trends according to the current locoregional situation.

## 1. Introduction

Urinary tract infections (UTIs) represent a common colonization followed by an inflammatory process of the urinary tract with various uropathogens. It is a widespread pathology, affecting over 150 million people around the globe annually, with over 10 million ambulatory visits and some 2 million emergency room visits estimated annually in the United States alone [1,2]. Hospitalizations for UTIs are expensive, costing almost USD 3 billion annually [3]. The growing resistance to routinely used antibiotics in both outpatient and inpatient settings adds to the burden of UTIs on global health systems [4]. These are considered the second-most common bacterial infections in humans, after respiratory infections, requiring special attention for a quick diagnosis and an optimal and effective treatment [5]. 

A UTI is one of the most common infections affecting women at different stages of life. According to estimates [6,7], every woman will have experienced at least one UTI over her lifetime, with more than 50% of all women experiencing a symptomatic UTI at least once [8]. Age raises the risk of infection [9]. Different types of conditions include lower UTIs (cystitis) and upper UTIs (pyelonephritis) [10]. Several risk factors are associated with acquiring UTIs and recurrent infections in female patients, such as facilitated ascent, which causes a higher incidence and a more significant public health problem than for the male population [11,12,13].

Antimicrobial resistance is a growing, global, and severe challenge to medical care. As a result, there are rising costs for patient care, an increase in hospital stays, and an increase in mortality. It has been found that almost all frequent infections in clinical practice exhibit high levels of resistance to conventional antibiotic treatments. It has also been reported that many organisms are multidrug-resistant [14]. 

The discovery of antibiotics represented a cornerstone in modern medicine, and they have served as a medication with obvious benefits on multiple infection sites. Still, misuse and abuse have also led to a severe hazard to public health because resistant uropathogenic bacteria are becoming more common. The European Association of Urology (EAU) Guidelines encourage the prudent use of all antimicrobials and propose Antibiotic Stewardship as a critical milestone in daily clinical practice. Furthermore, it suggests adapting antibiotic use policies to the local rates of resistance and sensitivity found, which will help both delay the increase in resistance and improve the efficacy of those indicated empirically [15]. Few recent data on local resistance rates are available for the Romanian territory [16,17,18], especially for more extended periods. The research that is currently available analyzes the bacterial prevalence and antibiotic resistance only for a determined period of a year [16,17,18]. The clinical practice requires reports that evaluate resistance rates at any particular time and indicate evolutionary trends across short and medium periods. The present study aims to assess the etiology and incidence of UTIs in Romania’s female population and evaluate the evolution of resistance and sensitivity rates to the common antibiotics over several periods for of years.

## 2. Results

The present study was developed in one of the largest urology hospitals in Bucharest, Romania—“Prof. Dr. Th. Burghele” Clinical Hospital—for 5 years. A total number of 1124 female patients met the study criteria. The evaluation was divided into three distinct periods of 4 months each with an interval of 2 years between them, as follows: 1 September 2018–31 December 2018—478 patients; 1 September 2020–31 December 2020—277 patients; 1 September 2022–31 December 2022—369 patients.

The most frequently encountered pathogen was *Escherichia coli*, representing 613 of the strains tested (54.53%), followed by *Klebsiella* spp., representing 186 of the strains tested (16.54%), *Proteus* spp., representing 81 strains (7.2%) and *Pseudomonas* spp., representing 42 strains (3.73%) in the group of Gram-negative bacteria. The most frequent Gram-positive bacteria tested were *Enterococcus* spp., with 164 strains tested (14.59%), followed by *Staphylococcus*, with 38 strains tested (3.38%). The same ratio between the tested uropathogens was maintained for all periods and age groups studied. We observed a linear increase in the incidence of each uropathogen that was directly proportional to the rise in the patient’s age. There are also minor exceptions to the rule. In the case of *Staphylococcus* spp., a higher incidence of 0.72% was observed in the age group 30–44 years, followed by the absence of detection in the immediately following age group, so that only in the > 60 years age group was it detected at a higher incidence of 2.16%. A detailed record of uropathogens’ incidence in all the periods studied and the division into age groups is presented in Table 1.

Considering *Escherichia coli* as the most frequently encountered uropathogen, it shows the overall highest rate of resistance among the studied antibiotics to levofloxacin R = 188 (30.66%), followed by amoxicillin-clavulanic ac R = 172 (28.05%), and ceftazidime R = 65 (10.6%). Maintained sensitivity was observed to fosfomycin 1 (0.16%), carbapenems—imipenem 3 (0.48%), meropenem 1 (0.16%), and nitrofurantoin 26 (4.24%).

There is a general tendency to increase this bacterial resistance to the tested antibiotics. The most alarming situations are found in the case of amikacin: 2018, R = 3.55% and 2022, R = 22.43%; amoxicillin-clavulanic acid: 2018, R = 21.34% and 2022, R = 37.07%; and ceftazidime: 2018, R = 7.11% and 2022, R = 17.07%. A decrease in resistance is observed for nitrofurantoin: 2018, R = 6.71%; 2020, R = 5.8%; and 2022, no resistance in the tested strains. A detailed evaluation of the resistance and sensitivity of *Escherichia coli* to the tested antibiotics in the studied periods is presented in Table 2.

Regarding *Klebsiella* spp., the second-most frequent pathogen studied, it shows the overall highest resistance to amoxicillin-clavulanic acid R = 68 (36.55%), ceftazidime R = 36 (19.35%), and levofloxacin R = 42 (22.58%). Relatively preserved sensitivity is observed for carbapenems—imipenem R = 7 (3.76%) and meropenem R = 9 (4.83%). The greatest increase in the evolution of resistance during the studied period was observed for levofloxacin 2018, R = 14.1% and 2022, R = 26.22%. A decrease in resistance was observed in the case of carbapenems imipenem 2018, R6.41% and 2022, R = 1.63%; meropenem: 2018, R = 6.41% and 2022, R = 4.91%; and ceftazidime: 2018, R = 20.51% and 2022, R = 18.03%. A detailed evaluation of the resistance and sensitivity of *Klebsiella* spp. to the tested antibiotics in the studied periods is presented in Table 3.

*Pseudomonas* spp. showed the highest resistance rates to all studied antibiotics throughout all evaluated periods. Alarming rates of resistance were registered for levofloxacin R = 23 (54.76%), ceftazidime R = 19 (45.23%), carbapenem—imipenem R = 17 (40.47%), and meropenem R = 16 (38.09%). It shows an overall increase in antibiotic resistance in all the tested antibiotics throughout the studied periods. A detailed evaluation of the resistance and sensitivity of *Pseudomonas* spp. to the tested antibiotics over different periods is presented in Table 4.

*Proteus* spp. displayed preserved rates of resistance to carbapenems—imipenem R = 4 (4.93%), meropenem R = 2 (2.46%), and ceftazidime R = 7 (8.64%). The highest resistance rates were observed for levofloxacin R = 30 (37.03%), followed by amoxicillin-clavulanic acid R = 25 (30.86%). Proteus demonstrated a decrease in resistance to all antibiotics studied throughout the evaluated periods in the study, except for levofloxacin, which showed an increase in resistance, as follows 2018, R = 32.35%, 2020, R = 34.61%, and 2022, R = 47.61%. A detailed evaluation of the resistance and sensitivity of *Proteus* spp. to the tested antibiotics in the studied periods is presented in Table 5.

The most common Gram-positive bacterium, *Enterococcus* spp., showed the highest overall resistance to levofloxacin, R = 75 (45.73%), followed by penicillin, R = 56 (34.14%) and ampicillin, R = 31 (18.9%). Reduced resistance rates are observed for vancomycin, R = 5 (3.04%), Fosfomycin, R = 5 (3.04%), and nitrofurantoin, R = 10 (6.09%). No strain studied in the evaluated periods demonstrated resistance to linezolid. Resistance increases are observed in the case of all antibiotics tested, except for penicillin: 2018, R = 34.52% vs. 2022, R = 30.9% and ampicillin: 2018, R = 20.23% vs. 2022, R = 10.9%. A detailed evaluation of the resistance and sensitivity of *Enterococcus* spp. to the tested antibiotics in the studied periods is represented in Table 6.

The least frequent uropathogen, *Staphylococcus* spp., showed the highest rate of resistance to penicillin, R = 20 (52.63%) and trimethoprim-sulfamethoxazole, R = 8 (21.05%). Maintained resistance rates were observed for nitrofurantoin R = 1 (2.63%) and linezolid R = 2 (5.6%). A decrease in the resistance of this uropathogen was observed in the case of trimethoprim-sulfamethoxazole and linezolid. A detailed evaluation of the resistance and sensitivity of *Staphylococcus* spp. to the tested antibiotics in the studied periods is represented in Table 7.

Considering all the Gram-negative strains’ overall resistance and sensitivity patterns, the highest resistance was observed for levofloxacin, followed by ceftazidime and amikacin. The highest sensitivity was observed for carbapenems—imipenem and meropenem. Considering the overall resistance and sensitivity patterns in all the Gram-positive strains, levofloxacin had the highest resistance, followed by penicillin. The most heightened sensitivity was observed for linezolid and nitrofurantoin. A visual representation of the overall resistance and sensitivity patterns in all the tested Gram-negative and positive strains is presented in Figure 1 and Figure 2.

The evolution of the resistance profiles of Gram-negative pathogens over the 5 years studied displayed a tendency to increase for all the usual antibiotics used to treat UTIs. The numbers observed were as follows: amikacin (2018, R = 5.8%; 2020, R = 5.76%; 2022, R = 18.33%), ceftazidime (2018, R = 11.8%; 2020, R = 13.58%; 2022 R = 17.33%), and levofloxacin (2018, R = 25.59%; 2020, R = 33.33%; 2022 R = 35.0%). Some constant evolution of carbapenem resistance was observed. A graphic visualization of the evolution of the Gram-negative uropathogens’ resistance to the tested antibiotics during all the studied periods is presented in Figure 3.

Considering the evolution of Gram-positive uropathogens’ resistance patterns, inconstant rates of resistance were detected. No linear evolution tendency was observed for any of the tested antibiotics. However, alarming resistance rates were observed for levofloxacin and penicillin. A tripling in the nitrofurantoin resistance rate was observed from 2018, R = 3.03% to 2022, R = 10.14%. Favorable resistance rates were observed for linezolid. A graphic visualization of the evolution of the resistance Gram-positive uropathogens to the tested antibiotics along all the studied periods is presented in Figure 4.

## 3. Discussion

The resistance of uropathogens to the conventional classes of antibiotics is a public health problem that has become more and more pronounced recently, and is considered to be one of the top most important risk factors for the safety of humanity. The increasingly aggressive prescription of antibiotics globally by health services, the empiric administration of first-line antibiotics in uncomplicated urinary infections, and the over-the-counter sale of these classes of drugs in many developing countries have led, in recent decades, to an overwhelming increase in resistance rates to the most common classes of antibiotics [19].

### 3.1. Differences and Trends Regarding the Prevalence of Uropathogens in Relation to the Patient’s Age

In the entire group of patients and in all the studied periods, progressive increases were observed, which were directly proportional between the subjects’ age and the incidence of uropathogens. A review of the literature published in the Aging Health Journal showed a higher incidence of UTI among women compared to men and a proportional increase with the age of the subjects, reaching over 10% per year among women over 60 and over 30% per year among women over 85 years old [20,21]. A recently published review [22] showed that 1–5% of healthy premenopausal women, 4–19% of otherwise healthy older women and men, and 15–50% of institutionalized elderly people experience asymptomatic bacteriuria, showing a similar linear increase in the incidence rates to our results. In the Netherlands, research on UTIs in people over the age of 85 found that women had a 1.7-fold higher risk than men (incidence of 12.8 per 100 persons each year; incidence of 7.8 per 100 persons each year) [23]. Women’s incidence of UTIs increased from 9 to 11% in subjects aged 65 to 74, from 11.4 to 14.3%, and from 14.7 to 19.8% in subjects aged 75 to 84 and > 85 years, respectively, according to a large observational study of UTIs in older adults conducted in the United Kingdom [24].

*Escherichia coli* is the most common ubiquitous uropathogen in the entire population studied, in all age groups, representing a total of 54.33% of all studied pathogens, which is significantly lower than those between 60% and 90% reported in countries such as Morocco [25], Portugal [26], or Pakistan [27]. Similarly, high rates of *Escherichia coli* incidence were detected in Western European countries, such as France or Austria, reporting rates over 65% in the studied populations [28]. The studies show similar incidence rates between 45 and 55% to those detected in the present survey in countries close to Romania, such as Hungary or Italy [29,30]. *Klebsiella* spp. was the second-most common Gram-negative uropathogen isolated in the studied cohort, with the incidence increasing linearly with each evaluated age group; similar results to those presented were found in studies in Libya [31] and Iraq [32]. In the present study, the most frequent Gram-positive uropathogen was presented by *Enterococcus* spp. with an almost 15% incidence rate, with the prevalence increasing linearly with the age group. However, other studies have shown significantly lower incidence rates of only 2.8% (Pakistan) [33] and 4.8% (Nepal) [34].

### 3.2. Evolution of the Resistance Patterns of Gram-Negative Uropathogens

A linearly increasing evolution was observed in the case of *Escherichia coli*, the most frequent Gram-negative pathogen throughout the studied periods in the case of all tested antibiotics, with few exceptions. The highest resistance rates were observed for levofloxacin (R = 30.66%) and amoxicillin-clavulanic acid (R = 28.05%). Similar data were noticed in a recent systematic review and metanalysis published by Ballesteros-Monrreal et al. [35], which showed an alarming increase in recent years of multidrug-resistant uropathogenic *Escherichia coli*, with a high incidence of aminopenicillins resistant strains—extended spectrum beta-lactamase (ESBL) representing over 55%. Another recent paper from the United States [36] highlighted the increasing incidence of fluoroquinolone-resistant strains of both *Escherichia coli* and *Klebsiella* spp., with alarmingly high rates of recent resistance, and underlined the urgent need for necessary measures to change the empirical treatment of UTIs.

Extensive research from Spain published in 2021 [37] on *Klebsiella* spp. isolated strains from different specimens underlined similarities in resistance trends regarding multiple antibiotic classes, and it especially highlighted the alarming resistance profiles in urine strain isolates compared with other sites, such as blood or respiratory infections. The highest rates of resistance among Gram-negative pathogens were observed in the case of *Pseudomonas* spp. Our results share similarities with recent data published in 2021 in a large study [38] about the patterns of this bacteria in UTIs; it highlighted a 27.7% resistance to amikacin, which is similar to the 30.95% resistance rates found in our study; 50% resistance to cephalosporins, which is close to the 45.23% resistance found in our research. Still, it showed 38.7% resistance to levofloxacin, which is considerably lower than the 54.76% resistance in the present study. *Proteus* spp., the most important uropathogen correlating UTIs and urinary lithiasis, was encountered in 7.2% of cases. The highest resistance was observed to levofloxacin (R = 37.05%), followed by amoxicillin–clavulanic ac. (R = 30.86%); these results show a better sensitivity compared to a recent publication from 2021 [39], which underlines high resistance to trimethoprim-sulfamethoxazole (R = 97%), nalidixic ac. (R = 93%), and amoxicillin (R = 62%) in the case of *Proteus* spp. A study from Hungary [40] that followed the evolution of Gram-negative pathogens’ resistance over 10 years reported high rates of multidrug-resistant *Proteus* spp. in the analyzed samples.

The evolution of the resistance of Gram-negative bacteria throughout the evaluated period is significant. It showed linear increases in amikacin, ceftazidime, and levofloxacin. Similar results were obtained by a recent study published last year [41], following over 12 years of observations at Peking University Hospital in Beijing, China. It highlighted high resistance in Gram-negative pathogens for cephalosporins and fluoroquinolones and alarmingly increasing carbapenem rates, especially in the geriatric population. It also concluded that the latter category is more susceptible to multidrug-resistant strains. Research published in April 2022, observing 10 years of the uropathogens’ resistance at the University of Gondar, Ethiopia [42], found more resistant uropathogens strains with an alarming evolution, represented by an overall resistance to *Escherichia coli* of R = 74% (amoxicillin-clavulanic ac.), R = 55.6% (ciprofloxacin), R = 24% (amikacin), and R = 26.5% (meropenem); *Klebsiella* spp.—R = 86.3% (amoxicillin-clavulanic ac.), R = 53.3% (ciprofloxacin), R = 36.4% (amikacin), and R = 17.6% (meropenem). The study highlighted that more than 44% of the total strains were multidrug-resistant during the observed period, emphasizing the severe evolution of pathogen resistance in the short and medium term of the observation [42].

The evolution of *Psudomonas* spp. resistance to various classes of antibiotics is alarming, both in terms of the increased frequency of strains’ resistant to at least one antibiotic and multidrug-resistant bacteria. A review from last year [43] shows, according to ECDC data, that 33.9% of all strains of *P. aeruginosa* in Europe are resistant to at least one of the studied antibiotics (cephalosporins, fluoroquinolones, and aminoglycosides). This increased resistance is observed especially in southern and eastern European countries, such as Greece, Bulgaria, Serbia, Slovakia, and Romania, where over 50% of the strains tested are resistant to at least one antibiotic from these classes, which is similar to the current study [43].

### 3.3. Evolution of the Resistance Patterns of Gram-Positive Uropathogens

*Enterococcus* spp. was the most frequent Gram-positive uropathogen, presenting high heterogenicity of resistance to different classes of antibiotics. The most significant rise in the resistance was observed for levofloxacin, from R = 38.09% (2018) to R = 52.78% (2022), and nitrofurantoin, from R = 3.57% (2018) to R = 10.9% (2022). A large study from Poland [44] followed the antibiotic susceptibility of *Enterococcus* strains in urinary probes from urology and nephrology patients and highlighted exciting results in terms of resistance patterns. Similar to our study, it presented high resistance to aminopenicillins and fluoroquinolones, with a resistance to norfloxacin between 50 and 80%, depending on the isolated strain. Ferede ZT et al. [45] showed no resistance to linezolid for all tested *Enterococci* strains, which is similar to our results. They noted 93.3% sensitivity to vancomycin, which is identical to our findings of 93.29%. In contrast, they observed extremely high resistance to ampicillin (80%), whereas our study shows only 18% overall resistance, with the highest sensitivity in the most recent determination—R = 10.9% (2022). Another publication analyzing the antibiotic resistance of *Enterococcus* spp. and its relation with biofilm formation [46] shows alarming data about resistance to different antibiotics. High resistance to levofloxacin (over 77%) and tetracyclin (over 86%) was encountered. Meanwhile, it shows relatively preserved sensitivity to nitrofurantoin, which is similar to our study and can still be a good option for treatment in uncomplicated cases. The presence of multidrug *Enterococcus* strains emerging is an alarming signal in recent publications, highlighting the presence of up to seven different classes of antibiotic resistance in selected strains, which shows an unfavorable evolution of the current resistance of this pathogen.

The least common Gram-positive uropathogen, *Staphylococcus* spp., presents relatively low rates of resistance evolution, except for penicillin (R = 52.63%) and trimetoprim-sulfamethoxazole (R = 21.05%). A recent study from 2022, following 1327 patients with UTIs from Tanzania [47], underlined alarming rates of resistance of this pathogen, which were much higher than our results; it shows over 40% resistance to nitrofurantoin, compared to 2.63% in our study, and linezolid over 18% resistance, compared to 5.25% in the current research. Another paper from France [48] shows relatively similar resistance rates to penicillin (R = 59.0%) and fluoroquinolones (R = 36.3%). Meanwhile, data from Southern Ireland at University Hospital Waterford [49] shows a significantly increased incidence of super-resistant *Staphylococcus aureus*, resulting in almost 28% of this pathogen’s total strains being methicillin-resistant. However, this research shows similar resistance to our study of nitrofurantoin (R = 2.7%), underlining its positive benefit in the treatment of Gram-positive UTIs.

### 3.4. The Implications of the COVID-19 Pandemic on the AMR of Uropathogens

The high antibiotic use among COVID-19 patients has amplified the antimicrobial resistance (AMR) issue. Antibiotics do not treat COVID-19; nonetheless, they are frequently administered in patients with respiratory disease due to early diagnostic ambiguity and worry about bacterial co-infection or subsequent infection in those who have confirmed COVID-19. In earlier evaluations [50,51,52,53], it was discovered that COVID-19 patients received a high percentage of antibiotic prescriptions (about 75%), despite only a small percentage of them having bacterial infections, especially those outside of the intensive care unit setting. One way to optimize antibiotic prescription and, at least in part, stop the emergence of antibiotic resistance is to be aware of local and regional antimicrobial susceptibility differences (AMR); thus, knowing local resistance evolution is essential.

According to recent data from 45 public and private clinics in Ireland, 76% of research participants stated that COVID-19 had a negative impact on how well antibiotic stewardship programs were implemented [54]. A study from Egypt published this year [55] following the evolution of AMR during the pandemic shows a significant increase in resistance for the vast majority of the tested strains. *Escherichia coli* showed an important rise in resistance to carbapenems, ceftazidime, and amikacin; *Klebsiella* to nitrofurantoin, gentamycin, and amoxicillin-clavulanic ac; and *Enterococcus* to trimethoprim-sulfamethoxazole, levofloxacin, and amikacin. A survey conducted by our team last year [18] studied resistance evolution pre-pandemic and during the pandemic, and showed a significant increase in resistance to fluoroquinolones and carbapenems of both *Klebsiella* spp. and *Pseudomonas*. It also highlighted the increased resistance of *Escherichia coli* to amoxicillin-clavulanic ac., levofloxacin, ceftazidime, and nitrofurantoin. Comparative research [56] that evaluated uropathogens’ antibiotic resistance changes in Iran during 2020 and 2022 highlighted an important resistance increase for ampicillin, carbapenems, and ceftazidime for *Escherichia coli*. Resistance rates were increased for *Klebsiella* spp. to ampicillin, levofloxacin, and ceftazidime. *Pseudomonas* spp. presented the lowest sensitivity to cefepime and carbapenems [56]. Similar research, published in February 2023 and conducted in Morocco [57], studied uropathogenic bacterial resistance profiles before and after the COVID-19 outbreak. It shows a significant increase in resistance, close to our results, especially for *Escherichia coli* to amoxicillin and levofloxacin; for *Klebsiella* spp. to amoxicillin and ceftriaxone; and for *Enterococcus* to levofloxacin and ciprofloxacin. Surprisingly, it showed a decreased resistance for *Klebsiella* spp. to amikacin, carbapenems, and trimethoprim-sulfamethoxazole, which is similar to our study, and for *Enterococcus* spp. to ceftriaxone, carbapenems, and trimethoprim-sulfamethoxazole.

Although the pandemic has had a worldwide impact, the negative repercussions are likely to be severe, leading to a higher burden of AMR. As a result, active antibiotic stewardship in all hospitals, clinics, and communities is required to ensure a sustainable future. As a result, concentrated global efforts and leadership with a higher level of public involvement and international cooperation are urgently needed to mitigate the pandemic’s negative impact on AMR.

### 3.5. Limitations

A few limitations have to be considered. The lack of additional information about patients’ medical histories, such as the consumption of antibiotics, history of urinary tract surgeries, or history of indwelling catheters—all critical contributors to the emergence and spread of resistant uropathogen strains—represents a limitation of this study. “There is strength in numbers”; thus, another limitation is the reduced quantity of analyzed urine cultures. The conclusions would be improved when the estimation of the evaluated probes was higher. However, this study presents information from female patients with a range of different pathologies from a major teaching hospital in the country’s capital city. Another fact that might influence our conclusions is the short time between the examined intervals, before, during, and immediately after the COVID-19 pandemic; the more time passes after the viral outbreak, the more trustworthy conclusions can be drawn.

More study is necessary to understand the dynamics of the viral illness in the evolution of antibiotic resistance in uropathogens. Despite the limitations, this study has the potential to advance knowledge about the fundamental role that the pandemic is playing in the selection of resistant strains of bacteria involved in UTIs and the emerging of AMR.

## 4. Materials and Methods

### 4.1. Study Design and Setting

The current cross-sectional retrospective study was conducted during three different periods of four months each, before, during, and immediately after the COVID-19 outbreak, between September 2018 and December 2022, as follows: 1 September 2018–31 December 2018; 1 September 2020–31 December 2020; and 1 September 2022–31 December 2022 at a major Urology Clinic—“Prof. Dr. Th. Burghele”—from Bucharest, Romania.

### 4.2. Study Population

As mentioned earlier, a total of 12,845 urine probes were analyzed during the period, of which 1124 samples met the inclusion criteria for this study. It involved only female patients over 18 years old, with positive urine culture—more than 10^5^ CFU/mL—and single bacterial strain on urine culture. The exclusion criteria of the current study were represented by male sex or patients under 18 years old, less than 10^5^ CFU/mL, two or more bacterial strains on urine culture, and patients with indwelling catheters. A representative flowchart of the study population is presented in Figure 5.

### 4.3. Data and Sample Collection

General information data were collected, such as age and sex for both hospitalized and non-hospitalized patients, while data regarding history, the behavioral, or clinical aspect of the selected study population were not able to be noted due to the retrospective character of the current research. After receiving adequate instructions, participants self-collected 5–10 mL of clean-catch, mid-stream urine (MSU) samples in a sterile urine container. In all cases, the collection of urine probes adhered to international safety guidelines [58]. Within 2 h after collection, samples were transferred to a designated box to the Clinic’s Microbiology Laboratories for further processing.

### 4.4. Quantitative Urine Culture, Bacterial Identification, and Antibiotic Susceptibility Test

After inoculating the urine samples with a sterile disposable loop on standard inoculation plates, they were incubated for 24 h. The bacteria were cultured on Columbia sheep agar and lactose agar using urine samples that had been obtained in a sterile container. In further instances, we cultured *Staphylococcus* spp. using the Chapman medium. On a culture media, significant microbial growth and colony morphology (e.g., color, size, and texture) and features were observed. Bacterial counts of more than 10^5^ CFU/mL of no more than two microorganism species were considered significant. Colony morphology, Gram stain, and many standard biochemical tests (lactose fermentation, catalase, oxidase, indole, methyl red, etc.) were used to identify the bacteria. Zones of antibiotic inhibition were interpreted per the Clinical and Laboratory Standards Institute (CLSI) recommendations for Antimicrobials Susceptibility Testing (AST), which was carried out using the Kirby–Bauer disk diffusion method [59]. Discussions of bacterial culture, uropathogen identification, and the implemented antibiotic susceptibility tests have already been reported in previous publications [14,16,17,18,60].

## 5. Conclusions

From all the urine samples analyzed throughout the evaluated periods, *Escherichia coli* and *Klebsiella* spp. are the most frequent Gram-negative uropathogens, while *Enterococcus* spp. is the usual Gram-positive bacteria involved in UTIs. In the case of all the pathogens, important variations in the resistance patterns of the different antibiotics tested were observed, following increases in resistance in most cases in the short and medium term.

We discovered the highest overall rates of resistance for *Pseudomonas* spp. There are also situations where the sensitivity increased throughout the evaluated period, but these were the exceptions. Briefly, a negative impact of the COVID-19 pandemic outbreak was observed in the evolution of the resistance rate of the main classes of antibiotics used in UTIs in all studied uropathogens.

Further studies are required to determine if this evolution is a phenomenon encountered in more regions and to take firm and rapid measures to slow down the process of AMR spreading fueled by antibiotic overprescription.

## Figures and Tables

**Figure 1 antibiotics-12-00948-f001:**
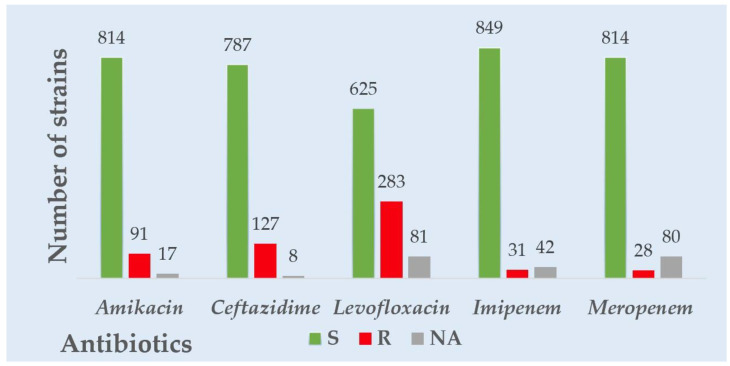
Overall Gram-negative strains’ resistance to common antibiotics tested.

**Figure 2 antibiotics-12-00948-f002:**
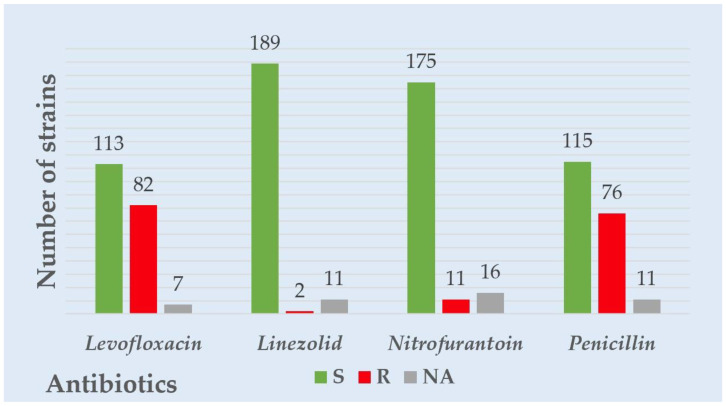
Overall Gram-positive strains’ resistance to common antibiotics tested.

**Figure 3 antibiotics-12-00948-f003:**
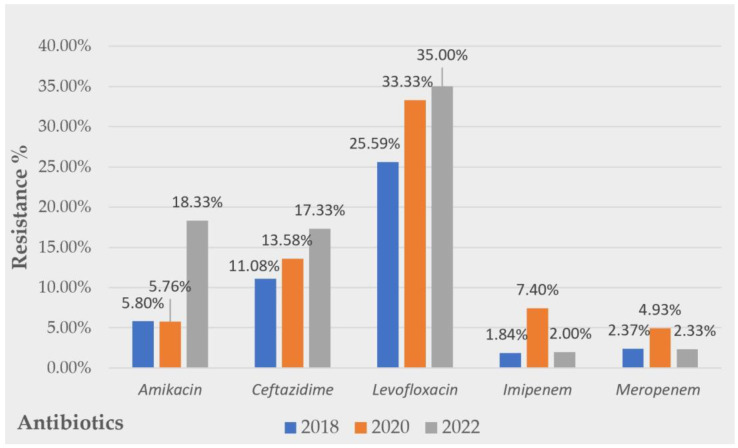
Evolution of Gram-negative strains’ resistance over a 5 year period to common antibiotics tested.

**Figure 4 antibiotics-12-00948-f004:**
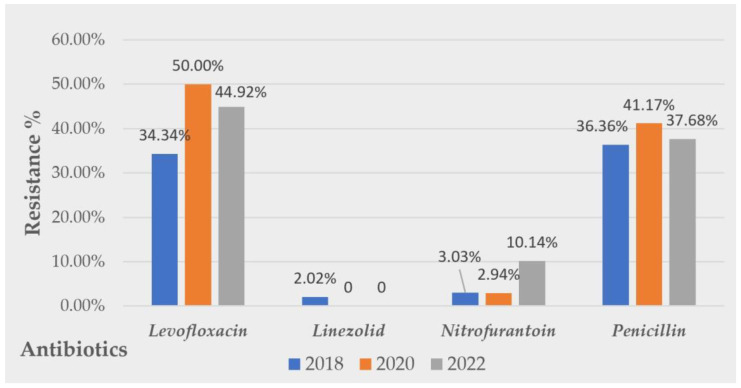
Evolution of Gram-positive strains’ resistance over 5 years to common antibiotics tested.

**Figure 5 antibiotics-12-00948-f005:**
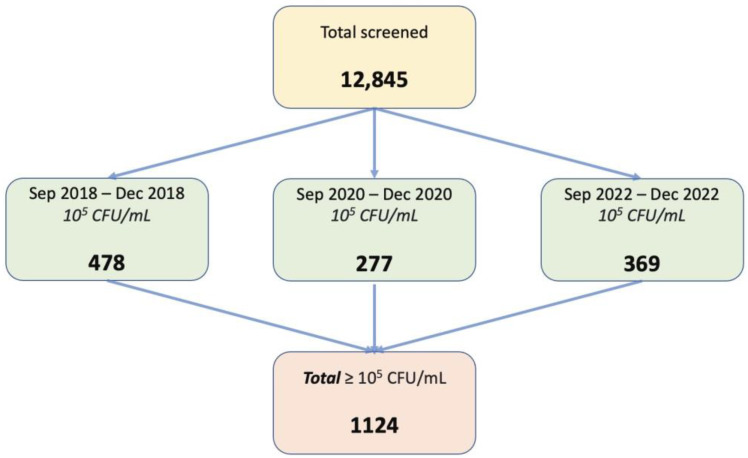
Flow-chart on patients’ distribution in the study.

**Table 1 antibiotics-12-00948-t001:** Uropathogens’ incidence and age stratifications.

Isolated Bacteria	19–29 Years			30–44 Years			45–59 Years			>60 Years			Total
	2018	2020	2022	2018	2020	2022	2018	2020	2022	2018	2020	2022	
	*n* (%)	*n* (%)	*n* (%)	*n* (%)	*n* (%)	*n* (%)	*n* (%)	*n* (%)	*n* (%)	*n* (%)	*n* (%)	*n* (%)	*n* (%)
** *Gram-negative* **	24 (5.02)	16 (5.77)	13 (3.52)	44 (9.2)	36 (12.99)	26 (7.04)	64 (13.38)	59 (21.29)	69 (18.69)	247 (51.67)	132 (47.65)	192 (52.03)	922 (82.02)
*Escherichia coli*	23 (4.81)	10 (3.61)	8 (2.16)	30 (6.27)	25 (9.02)	17 (4.6)	45 (9.41)	35 (12.63)	44 (11.92)	155 (32.42)	85 (30.68)	136 (36.85)	613 (54.53)
*Klebsiella* spp.	0	3 (1.08)	3 (0.81)	10 (2.09)	6 (2.16)	8 (2.16)	11 (2.3)	14 (5.05)	18 (4.87)	57 (11.92)	24 (8.66)	32 (8.67)	186 (16.54)
*Proteus* spp.	0	3 (1.08)	2 (0.54)	3 (0.62)	5 (1.8)	1 (0.27)	7 (1.46)	4 (1.44)	4 (1.08)	24 (5.02)	14 (5.05)	14 (3.79)	81 (7.2)
*Pseudomonas* spp.	1 (0.2)	0	0	1 (0.2)	0	0	1 (0.2)	6 (2.16)	3 (0.81)	11 (2.3)	9 (3.24)	10 (2.71)	42 (3.73)
** *Gram-positive* **	5 (1.04)	2 (7.22)	7 (1.89)	16 (3.34)	4 (1.44)	6 (1.62)	21 (4.39)	5 (1.8)	17 (4.6)	57 (11.92)	23 (8.3)	39 (10.56)	202 (17.97)
*Enterococcus* spp.	5 (1.04)	1 (0.36)	5(1.35)	12 (2.51)	2 (0.72)	3 (0.81)	18 (3.76)	5 (1.8)	13 (3.52)	49 (10.25)	17 (6.13)	34 (9.21)	164 (14.59)
*Staphylococcus* spp.	0	1 (0.36)	2 (0.54)	4 (0.83)	2 (0.72)	3 (0.81)	3 (0.62)	0	4 (1.08)	8 (1.67)	6 (2.16)	5 (1.35)	38 (3.38)
**Total**	29 (6.06)	18 (6.49)	20 (5.42)	60 (12.55)	40 (14.44)	32 (8.67)	85 (17.78)	64 (23.1)	86 (23.3)	304 (63.59)	155 (55.95)	231 (62.6)	1124

*n*—number; %—percentage.

**Table 2 antibiotics-12-00948-t002:** *Escherichia coli* susceptibility and resistance patterns.

Tested Antibiotics	2018			2020			2022			Total		
	S	R	NA	S	R	NA	S	R	NA	S	R	NA
	*n* (%)	*n* (%)	*n* (%)	*n* (%)	*n* (%)	*n* (%)	*n* (%)	*n* (%)	*n* (%)	*n* (%)	*n* (%)	*n* (%)
Amikacin	244 (96.44)	9 (3.55)	–	148 (95.48)	3 (1.93)	4 (2.58)	152 (74.14)	46 (22.43)	7 (3.41)	544 (88.74)	58 (9.46)	11 (1.79)
Amoxicillin— Clavulanic ac.	198 (78.26)	54 (21.34)	1 (0.39)	107 (69.03)	42 (27.09)	6 (3.87)	127 (61.95)	76 (37.07)	2 (0.97)	432 (70.47)	172 (28.05)	9 (1.46)
Ceftazidime	232 (91.69)	18 (7.11)	3 (1.18)	143 (92.25)	12 (7.74)	–	170 (82.92)	35 (17.07)	–	545 (88.9)	65 (10.6)	3 (4.89)
Fosfomycin	226 (89.23)	–	27 (10.67)	143 (92.25)	1 (0.64)	11 (7.09)	201 (98.04)	–	4 (1.95)	570 (92.98)	1 (0.16)	42 (6.85)
Imipenem	249 (98.41)	–	4 (1.58)	148 (95.48)	2 (1.29)	5 (3.22)	187 (91.21)	1 (0.48)	17 (8.29)	584 (95.26)	3 (0.48)	26 (4.24)
Levofloxacin	176 (69.56)	72 (28.45)	5 (1.97)	110 (70.96)	43 (27.74)	2 (1.29)	131 (63.9)	73 (35.6)	1 (4.87)	417 (68.02)	188 (30.66)	8 (1.3)
Meropenem	248 (98.02)	1 (0.39)	4 (1.58)	151 (97.41)	–	4 (2.58)	155 (75.6)	–	50 (24.93)	554 (90.37)	1 (0.16)	58 (9.46)
Nitrofurantoin	148 (58.49)	17 (6.71)	88 (34.78)	110 (70.96)	9 (5.8)	36 (23.22)	150 (73.17)	–	55 (26.82)	408 (66.55)	26 (4.24)	179 (29.2)

*n*—number; %—percentage; R—resistance; S—susceptibility; NA—not available.

**Table 3 antibiotics-12-00948-t003:** *Klebsiella* spp. susceptibility and resistance patterns.

Tested Antibiotics	2018			2020			2022			Total		
	S	R	NA	S	R	NA	S	R	NA	S	R	NA
	*n* (%)	*n* (%)	*n* (%)	*n* (%)	*n* (%)	*n* (%)	*n* (%)	*n* (%)	*n* (%)	*n* (%)	*n* (%)	*n* (%)
Amikacin	71 (91.02)	7 (8.97)	-	42 (89.36)	3 (6.38)	2 (4.25)	56 (91.8)	5 (9.8)	-	169 (90.86)	15 (8.06)	2 (1.07)
Amoxicillin— Clavulanic ac.	50 (4.1)	27 (34.61)	1 (1.28)	25 (53.19)	18 (38.29)	4 (8.51)	36 (59.01)	23 (37.7)	2 (3.27)	111 (59.67)	68 (36.55)	7 (3.76)
Ceftazidime	60 (76.92)	16 (20.51)	2 (2.56)	37 (78.72)	9 (19.14)	1 (2.12)	50 (81.96)	11 (18.03)	-	147 (79.03)	36 (19.35)	3 (1.61)
Imipenem	72 (92.3)	5 (6.41)	1 (1.28)	43 (91.8)	1 (2.12)	3 (6.38)	55 (90.16)	1 (1.63)	5 (8.19)	170 (91.39)	7 (3.76)	9 (4.83)
Levofloxacin	66 (84.61)	11 (14.1)	1 (1.28)	30 (63.82)	15 (31.91)	2 (4.25)	44 (72.13)	16 (26.22)	1 (1.63)	140 (75.26)	42 (22.58)	4 (2.15)
Meropenem	72 (2.3)	5 (6.41)	1 (1.28)	43 (91.48)	1 (2.12)	3 (6.38)	50 (81.96)	3 (4.91)	8 (13.11)	165 (88.7)	9 (4.83)	12 (6.45)

*n*—number; %—percentage; R—resistance; S—susceptibility; NA—not available.

**Table 4 antibiotics-12-00948-t004:** *Pseudomonas* spp. susceptibility and resistance patterns.

Tested Antibiotics	2018			2020			2022			Total		
	S	R	NA	S	R	NA	S	R	NA	S	R	NA
	*n* (%)	*n* (%)	*n* (%)	*n* (%)	*n* (%)	*n* (%)	*n* (%)	*n* (%)	*n* (%)	*n* (%)	*n* (%)	*n* (%)
Amikacin	12 (85.71)	2 (14.28)	-	7 (46.66)	7 (46.66)	1 (0.66)	8 (61.53)	4 (30.76)	1 (7.69)	27 (64.28)	13 (30.95)	2 (4.76)
Ceftazidime	10 (71.42)	3 (21.42)	1 (7.14)	4 (26.66)	11 (73.33)	-	7 (53.84)	5 (38.46)	1 (7.69)	21 (50.0)	19 (45.23)	2 (4.76)
Imipenem	11 (78.57)	2 (14.28)	1 (7.14)	4 (26.66)	11 (73.33)	-	8 (61.53)	4 (30.76)	1 (7.69)	23 (54.76)	17 (40.47)	2 (4.76)
Levofloxacin	11 (78.57)	3 (21.42)	-	1 (0.66)	14 (93.33)	-	7 (53.84)	6 (46.15)	-	19 (45.23)	23 (54.76)	-
Meropenem	11 (78.57)	2 (14.28)	1 (7.14)	4 (26.66)	10 (66.66)	1 (0.66)	9 (69.23)	4 (30.76)	-	24 (57.14)	16 (38.09)	2 (4.76)

*n*—number; %—percentage; R—resistance; S—susceptibility; NA—not available.

**Table 5 antibiotics-12-00948-t005:** *Proteus* spp. susceptibility and resistance patterns.

Tested Antibiotics	2018			2020			2022			Total		
	S	R	NA	S	R	NA	S	R	NA	S	R	NA
	*n* (%)	*n* (%)	*n* (%)	*n* (%)	*n* (%)	*n* (%)	*n* (%)	*n* (%)	*n* (%)	*n* (%)	*n* (%)	*n* (%)
Amikacin	30 (88.23)	4 (11.76)	-	23 (88.46)	1 (3.84)	2 (7.69)	21 (100.0)	-	-	74 (91.35)	5 (6.17)	2 (2.46)
Amoxicillin— Clavulanic ac.	20 (58.82)	11 (32.35)	3 (8.82%)	11 (24.3)	8 (30.76)	7 (26.92)	15 (71.42)	6 (28.57)	-	46 (56.79)	25 (30.86)	10 (12.34)
Ceftazidime	29 (85.29)	5 (14.7)	-	25 (96.15)	1 (3.84)	-	20 (95.43)	1 (4.76)	-	74 (91.35)	7 (8.64)	-
Imipenem	33 (97.05)	-	1 (2.94%)	20 (76.92)	4 (15.38)	2 (7.69)	19 (90.47)	-	2 (9.52)	72 (88.88)	4 (4.93)	5 (6.17)
Levofloxacin	21 (61.76)	11 (32.35)	2 (5.88%)	17 (65.38)	9 (34.61)	-	11 (52.38)	10 (47.61)	-	49 (60.49)	30 (37.03)	2 (2.46)
Meropenem	31 (91.17)	1 (2.94)	2 (5.88)	24 (92.3)	1 (3.84)	1 (3.84)	16 (76.19)	-	5 (23.8)	71 (87.65)	2 (2.46)	8 (9.87)

*n*—number; %—percentage; R—resistance; S—susceptibility; NA—not available.

**Table 6 antibiotics-12-00948-t006:** *Enterococcus* spp. susceptibility and resistance patterns.

Tested Antibiotics	2018			2020			2022			Total		
	S	R	NA	S	R	NA	S	R	NA	S	R	NA
	*n* (%)	*n* (%)	*n* (%)	*n* (%)	*n* (%)	*n* (%)	*n* (%)	*n* (%)	*n* (%)	*n* (%)	*n* (%)	*n* (%)
Ampicillin	60 (71.42)	17 (20.23)	7 (8.33)	15 (60.0)	8 (32.0)	2 (8.0)	48 (87.27)	6 (10.9)	1 (1.81)	123 (75.0)	31 (18.9)	10 (6.09)
Fosfomycin	78 (92.85)	1 (1.19)	5 (5.95)	22 (88.0)	2 (8.0)	1 (4.0)	53 (96.36)	2 (3.63)	-	153 (93.29)	5 (3.04)	6 (3.65)
Levofloxacin	51 (60.71)	32 (38.09)	1 (1.19)	8 (32.0)	14 (56.0)	3 (12.0)	25 (45.45)	29 (52.72)	1 (1.81)	84 (51.21)	75 (45.73)	5 (3.04)
Linezolid	77 (91.66)	-	7 (8.33)	24 (96.0)	-	1 (4.0)	54 (98.18)	-	1 (1.81)	155 (94.51)	-	9 (5.48)
Nitrofurantoin	76 (90.47)	3 (3.57)	5 (5.95)	24 (96.0)	1 (4.0)	-	47 (85.45)	6 (10.9)	2 (3.63)	147 (89.63)	10 (6.09)	7 (4.26)
Penicillin	46 (54.76)	29 (34.52)	9 (10.71)	14 (56.0)	10 (40.0)	1 (4.0)	38 (69.09)	17 (30.9)	-	98 (59.75)	56 (34.14)	10 (6.09)
Vancomycin	79 (94.04)	-	5 (5.95)	25 (100.0)	-	-	49 (89.09)	5 (9.09)	1 (1.81)	153 (93.29)	5 (3.04)	6 (3.65)

*n*—number; %—percentage; R—resistance; S—susceptibility; NA—not available.

**Table 7 antibiotics-12-00948-t007:** *Staphylococcus* spp. sensitivity and resistance patterns.

Tested Antibiotics	2018			2020			2022			Total		
	S	R	NA	S	R	NA	S	R	NA	S	R	NA
	*n* (%)	*n* (%)	*n* (%)	*n* (%)	*n* (%)	*n* (%)	*n* (%)	*n* (%)	*n* (%)	*n* (%)	*n* (%)	*n* (%)
Trimethoprim— Sulfamethoxazole	9 (60.0)	4 (26.66)	2 (13.33)	8 (88.88)	1 (11.11)	-	10 (71.42)	3 (21.42)	1 (7.14)	27 (71.05)	8 (21.05)	3 (7.89)
Levofloxacin	11 (73.33)	2 (13.33)	2 (13.33)	6 (66.66)	3 (33.33)	-	12 (85.71)	2 (14.28)	-	29 (76.31)	7 (18.42)	2 (5.26)
Linezolid	12 (80.0)	2 (13.3)	1 (6.66)	8 (88.88)	-	1 (11.11)	14 (100.0)	-	-	34 (89.47)	2 (5.26)	2 (5.26)
Nitrofurantoin	13 (86.66)	-	2 (13.33)	7 (77.77)	-	2 (22.22)	8 (57.14)	1 (7.14)	5 (35.71)	28 (73.68)	1 (2.63)	9 (23.68)
Penicillin	8 (53.33)	7 (46.66)	-	5 (55.55)	4 (44.44)	-	4 (28.57)	9 (64.28)	1 (7.14)	17 (44.73)	20 (52.63)	1 (2.63)

*n*—number; %—percentage; R—resistance; S—susceptibility; NA—not available.

## Data Availability

Data supporting the reported results are available from the authors.

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
