# Peer review of "Uropathogens’ Antibiotic Resistance Evolution in a Female Population: A Sequential Multi-Year Comparative Analysis"

_antibiotics, 2023, doi:10.3390/antibiotics12060948_

Round 1

Reviewer 1 Report

The study done by MareÈ™ et al represent retrospective analysis regarding incidence of uropathogens and the evolution of resistance rates. The findings of this study is alarming the community about the antibiotic misuse in case of viral infections especially during the pandemic. The authors have found the highest overall resistance in Gram-negative uropathogens was observed for Levofloxacin followed by  Ceftazidime and Amikacin, while the highest resistance in Gram-positive uropathogens was observed for Levofloxacin, and Penicillin. Regarding the evolution of resistance pattern by time ,a linear evolution was observed for Ceftazidime, and Levofloxacin for Gram-negative uropathogens. The study is very interesting and have great merit, the manuscript is well written, the results are clearly represented. Some minor errors have to be corrected:

1- Type all bacterial names in the abstract section italic

2- line 423: replace 105 to 10 to power 5

3- Fig.1 and Fig. 2: type "R" in the legend instead of "negative"

4- I suggest to correlate the title of the manuscript to COVID and include it as well in the keywords, since the main objective of the study is to assess the impact of COVID pandemic on the evolution of AMR

Reviewer 2 Report

The reason for authors using the word “germs” may be inappropriate for this journal.

Line #78: The evaluation was divided into 3 distinct periods of 4 months each with an interval of 2 years between them, as follows: 1st September 2018 80 - 31st December 2018 - 478 patients, 1st September 2020 - 31st December 2020 - 277 patients, 1st September 2022 - 31st December 2022 - 369 patients. The authors did not give a justification for the selected periods of the 3 years. Why not 12 months of each year and instead choose 4/12 months for each of these years?

1124 positive urine cultures…authors should include the total number of samples that were tested.

Line #136: No strain studied in the evaluated periods demonstrated resistance to linezolid but then line #144 shows a positive correlation in linezolid R=2 (5.6%)  | Maintained resistance rates were observed for nitrofurantoin R=1 (2.63%) and linezolid  R=2 (5.6%). Also from Figure 2. Overall Gram-positive strains` resistance to common antibiotics tested in line 185 shows two isolates being resistant to linezolid.

Reviewer 3 Report

Dear Authors, 

the review of your article is as follow:

In the article titled “Uropathogens’ Antibiotic Resistance Evolution in a Female Population: a Sequential muti-year camparative Analysis” the authors presented the importance of evolving antibiotic resistance in bacterial strains that cause UTIs in female Romanian patients. General commentary All microorganism names should be written in italics. All antibiotic names should be written in lower case. Line 86: replace the word “germ” with the word “pathogen” The word “represneted” (e.g. line 94 and onward) should be changed to “presented”. Line 113, 114, 115: missing coma after the “R=x%” data Table 1: the “Coli” of Escherichia coli should be written in lower case. Line 147: Misspelled name of Staphylococcus (written “staphilococcus”) Line 294: According to “ECDC” not “eCDC” Major concerns Line 35: “Locoregional situation” is an unclear term. Line 39: The sentence should be re-written. Urinary Tract Infections are not the same as a “common colonization”. A colonization does not involved an inflammatory process while an infection does. Section 1, introduction This section is lacking data about the epidemiology of UTIs in the EU as well as in Romanian. It is unclear to the reviewer why the authors decided to use statistics from North America (the U.S. in particular) to compare against when the study’s materials were collected from a region that is culturally and geographically different. This potentially opens the study up to flaws in the conclusions drawn from the data. Line 70, 71: Must be elaborated. Line 73: Aim of the study does not correspond with the data pool that has been collected (considering that the survey is based on microbiologic data from a single hospital). Another major concern of the reviewer is the lack is identification of bacteria down to the specific species since the antibiotic profile of different pathogens can vary immensely depending on the species (e.g. Enteroccocus faecium vs Enterococcus fecalis). This sort of imprecision prevents the interpretation of critical portions of data (e.g. Did the ratio of Enterococcus faecium/Enterococcus fecalis infections change, or has resistance to ampicillin increased in the Enteroccus fecalis population). Another major concern is the lack of identification of specific mechanisms of resistance in the pathogens that have been found to be resistant to antimicrobial agents (e.g. line 111: Klebsiella pneumonia resistance to carbapenems). This same concern applies to Pseudomonas (line 118). Another example of imprecise identification applies to the Staphylococci. Throughout the study there is no differentiation between the coagulase positive and coagulase negative staphylococci. In Table 7, there is a lack of information regarding the susceptibility to cefoxitin, and therefore the inability to detect methicillin resistant staphylococci. In Table 2 (and onward), please explain what “Not applicable” susceptibility means. In certain pathogens there is a number given in the column, and in certain pathogens a “-” in given in the column. In table legends and table titles and figure titles, change “sensitivity” to “susceptibility”. The reviewer is concerned that despite the fact that Romanian is part of the EU, the authors did not use EU standards (EUCAST system) for identifying antimicrobial resistance, rather opting for American standards (ECDC). Materials and methods should be presented before results. Figure 5 (flowchart) should include not only the number of patients samples in the study, but also the number of cultures, number of excluded samples with the reason for exclusion. In the present form, the flow chart does not add any useful information, and could be simply removed from the article altogether. An additional shortcoming of the article is its lack of statistical analysis of trends found in the data. The reviewer kinda requests the authors to justify why only the September-December period was used between 2018-2022 rather than any other period of the year. The reviewer also has concerns about the fact that over 50% of the references used for this article are older than 2020, and therefore their relevance to current resistance profiles. E.g. reference 26, which describes UTIs in madagascar, published in 2007. References 27, which describes UTIs in Rawanda from 2011.

Round 2

Reviewer 3 Report

Dear Authors, 

for your responce to my review, all explanations and modification.